| $5.24_{\pm0.69}$ | $100_{\pm0.00}$ | $94.26_{\pm0.02}$ | $12.88_{\pm0.09}$ | 0.00 | 44.56 |
| FT | $0.63_{\pm4.61}$ | $99.88_{\pm0.12}$ | $94.06_{\pm0.20}$ | $2.70_{\pm10.19}$ | 3.78 | 2.45 |
| RL | $7.61_{\pm2.37}$ | $99.67_{\pm0.33}$ | $92.83_{\pm1.43}$ | $37.36_{\pm24.47}$ | 7.15 | 2.73 |
| GA | $0.69_{\pm4.56}$ | $99.50_{\pm0.50}$ | $94.01_{\pm0.25}$ | $1.70_{\pm11.18}$ | 4.12 | 0.15 |
| IU | $1.07_{\pm4.17}$ | $99.20_{\pm0.80}$ | $93.20_{\pm1.06}$ | $2.67_{\pm10.21}$ | 4.06 | 0.39 |
| BE | $0.59_{\pm4.65}$ | $99.42_{\pm0.58}$ | $93.85_{\pm0.42}$ | $7.47_{\pm5.41}$ | 2.76 | 0.27 |
| BS | $1.78_{\pm3.47}$ | $98.29_{\pm1.71}$ | $92.69_{\pm1.57}$ | $8.96_{\pm3.93}$ | 2.67 | 0.45 |
| $\ell_1$-sparse | $4.19_{\pm1.06}$ | $97.74_{\pm2.26}$ | $91.59_{\pm2.67}$ | $9.84_{\pm3.04}$ | 2.26 | 2.48 |
| SalUn | $2.85_{\pm2.39}$ | $99.62_{\pm0.38}$ | $93.93_{\pm0.33}$ | $14.39_{\pm1.51}$ | 1.15 | 2.74 |
| Unlink[†] | $1.49_{\pm0.12}$ | $98.89_{\pm0.44}$ | $92.76_{\pm0.23}$ | $7.87_{\pm0.11}$ | 2.84 | 0.42 |

## F.3 MULTI-CLASS UNLEARNING ON CIFAR-100

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

| Methods | UA↑ | RA↑ | TA↑ | MIA↑ | Avg.Gap↓ | RTE (min.)↓ |
|---|---|---|---|---|---|---|
| Retrain | 100.00 | 95.41 | 80.85 | 100.00 | - | 62.69 |
| FT | $92.56_{\pm7.28}$ | $89.66_{\pm0.98}$ | $79.28_{\pm1.34}$ | $95.18_{\pm5.73}$ | 4.90 | 4.10 |
| IU | $74.64_{\pm24.20}$ | $70.36_{\pm29.11}$ | $60.86_{\pm23.68}$ | $69.95_{\pm31.08}$ | 25.11 | 1.19 |
| BE | $98.35_{\pm0.84}$ | $79.71_{\pm4.82}$ | $61.35_{\pm3.62}$ | $98.16_{\pm0.10}$ | 8.05 | 0.44 |
| BS | $97.99_{\pm5.12}$ | $83.07_{\pm6.76}$ | $65.21_{\pm5.05}$ | $99.01_{\pm2.00}$ | 6.10 | 0.87 |
| $\ell_1$-sparse | $96.30_{\pm5.16}$ | $87.88_{\pm1.18}$ | $78.66_{\pm1.58}$ | $97.57_{\pm4.19}$ | 3.96 | 4.17 |
| SalUn | $99.99_{\pm0.03}$ | $94.51_{\pm0.44}$ | $81.44_{\pm1.27}$ | $100.00_{\pm0.00}$ | 0.37 | 4.41 |
| SSD | $98.17_{\pm2.43}$ | $88.35_{\pm5.10}$ | $76.32_{\pm3.55}$ | $99.56_{\pm0.75}$ | 3.46 | 0.51 |
| GF | $94.14_{\pm5.85}$ | $83.93_{\pm17.17}$ | $64.42_{\pm13.09}$ | $95.17_{\pm3.71}$ | 9.65 | 1.24 |
| Unlink | $99.93_{\pm0.10}$ | $96.06_{\pm0.30}$ | $80.65_{\pm1.01}$ | $100.00_{\pm0.00}$ | **0.23** | **0.01** |

Table 15: Results of class-wise forgetting on ResNet18.

| Dataset | Methods | UA↑ | RA↑ | TA↑ | MIA↑ | Avg.Gap↓ | RTE (min.)↓ |
|---|---|---|---|---|---|---|---|
| | Retrain | 100.00 | 100.00 | 94.69 | 100.00 | - | 35.65 |
| CIFAR-10 | FT | $100.00_{\pm0.00}$ | $90.43_{\pm2.47}$ | $86.36_{\pm2.32}$ | $100.00_{\pm0.00}$ | 4.47 | 2.29 |
| | GA | $93.63_{\pm1.54}$ | $94.21_{\pm1.91}$ | $88.43_{\pm01.94}$ | $96.38_{\pm1.93}$ | 5.51 | 0.14 |
| | IU | $91.63_{\pm12.20}$ | $84.77_{\pm24.73}$ | $79.79_{\pm22.97}$ | $85.14_{\pm7.51}$ | 13.33 | 0.39 |
| | BE | $83.57_{\pm4.10}$ | $98.44_{\pm0.47}$ | $92.62_{\pm1.06}$ | $99.26_{\pm0.70}$ | 5.19 | 0.28 |
| | BS | $85.24_{\pm11.48}$ | $98.03_{\pm1.03}$ | $92.21_{\pm1.69}$ | $98.72_{\pm1.13}$ | 5.12 | 0.50 |
| | $\ell_1$-sparse | $100.00_{\pm0.00}$ | $97.49_{\pm0.54}$ | $91.79_{\pm0.88}$ | $100.00_{\pm0.00}$ | 1.35 | 2.36 |
| | SalUn | $99.95_{\pm0.15}$ | $99.78_{\pm0.09}$ | $94.37_{\pm0.68}$ | $100.00_{\pm0.00}$ | **0.15** | 2.45 |
| | SSD | $100.00_{\pm0.00}$ | $98.21_{\pm1.85}$ | $92.84_{\pm1.98}$ | $100.00_{\pm0.00}$ | 0.91 | 0.21 |
| | GF | $94.14_{\pm8.80}$ | $89.25_{\pm7.17}$ | $84.18_{\pm6.68}$ | $98.21_{\pm4.16}$ | 7.22 | 0.41 |
| | Unlink | $98.04_{\pm0.62}$ | $99.47_{\pm0.06}$ | $94.91_{\pm0.60}$ | $100.00_{\pm0.00}$ | 0.67 | **0.01** |
| | Retrain | 100.00 | 100.00 | 95.97 | 100.00 | - | 43.16 |
| SVHN | FT | $100.00_{\pm0.00}$ | $98.19_{\pm0.39}$ | $92.46_{\pm0.61}$ | $100.00_{\pm0.00}$ | 1.32 | 2.65 |
| | GA | $97.56_{\pm2.34}$ | $98.38_{\pm0.91}$ | $93.45_{\pm0.78}$ | $98.95_{\pm2.26}$ | 1.90 | 0.16 |
| | IU | $90.70_{\pm21.34}$ | $98.89_{\pm1.42}$ | $94.21_{\pm1.82}$ | $99.96_{\pm0.11}$ | 3.04 | 0.44 |
| | BE | $98.29_{\pm0.07}$ | $99.55_{\pm0.10}$ | $94.92_{\pm1.12}$ | $100.00_{\pm0.00}$ | 0.80 | 0.32 |
| | BS | $85.09_{\pm11.95}$ | $99.36_{\pm0.11}$ | $94.07_{\pm0.66}$ | $91.03_{\pm11.20}$ | 6.60 | 0.57 |
| | $\ell_1$-sparse | $99.56_{\pm0.00}$ | $99.16_{\pm0.13}$ | $94.11_{\pm0.41}$ | $100.00_{\pm0.00}$ | 0.78 | 2.69 |
| | SalUn | $99.93_{\pm0.08}$ | $99.99_{\pm0.00}$ | $95.99_{\pm0.14}$ | $100.00_{\pm0.00}$ | **0.02** | 2.87 |
| | SSD | $100.00_{\pm0.00}$ | $97.37_{\pm4.18}$ | $91.90_{\pm5.19}$ | $100.00_{\pm0.00}$ | 1.67 | 0.24 |
| | GF | $91.17_{\pm19.02}$ | $98.51_{\pm0.64}$ | $93.81_{\pm0.86}$ | $100.00_{\pm0.00}$ | 3.12 | 0.41 |
| | Unlink | $98.59_{\pm0.73}$ | $99.43_{\pm0.17}$ | $95.06_{\pm0.51}$ | $100.00_{\pm0.00}$ | 0.72 | **0.01** |

the singular values to select these vectors which are less than $\beta$. Table 10 shows the results of 10% random forgetting on ResNet18 trained on CIFAR-10. Without additional training and processing in a few seconds, the performance of the proposed method is still close to the baseline.

# G  MORE VISUALIZATION

Figure 4 shows more generative results of class-wise forgetting for Stable Diffusion on the Imagenette dataset. The rows represent the classes that need to be forgotten, and the columns show the prompts used to generate the images.

# H  ERASURE IN CONVOLUTION.

While convolutional layers operate differently from fully connected layers, their operations can be reformulated as matrix multiplications, allowing the proposed unlearning method for fully connected

Table 16: Results of class-wise forgetting on ResNet50.

| Dataset | Methods | UA↑ | RA↑ | TA↑ | MIA↑ | Avg.Gap↓ | RTE (Mins)↓ |
|---------|---------|-----|-----|-----|------|----------|-------------|
| | Retrain | 100.00 | 99.99 | 94.19 | 100.00 | - | 88.42 |
| | FT | 98.82 | 97.54 | 91.86 | 100.00 | 1.48 | 5.52 |
| | GA | 95.46 | 90.54 | 85.32 | 96.55 | 6.57 | 0.33 |
| | IU | 78.52 | 91.11 | 85.86 | 84.47 | 13.55 | 1.01 |
| CIFAR-10 | BE | 77.97 | 96.60 | 75.86 | 90.47 | 8.64 | 0.63 |
| | BS | 77.68 | 96.49 | 90.47 | 93.08 | 9.11 | 1.26 |
| | $\ell_1$-sparse | 100.00 | 94.91 | 90.32 | 100.00 | 2.23 | 5.63 |
| | SalUn | 100.00 | 99.15 | 93.61 | 100.00 | **0.35** | 6.11 |
| | Unlink | 97.56 | 99.47 | 94.85 | 100.00 | 0.89 | **0.02** |
| | Retrain | 100.00 | 99.93 | 74.19 | 100.00 | - | 97.37 |
| | FT | 95.71 | 93.57 | 68.51 | 99.77 | 4.08 | 6.11 |
| | GA | 77.44 | 93.25 | 68.60 | 90/78 | 11.01 | 0.04 |
| | IU | 95.75 | 75.62 | 57.03 | 98.84 | 11.72 | 0.82 |
| CIFAR-100 | BE | 94.27 | 86.33 | 63.49 | 97.53 | 8.12 | 0.08 |
| | BS | 94.04 | 86.39 | 63.56 | 97.22 | 8.23 | 0.14 |
| | $\ell_1$-sparse | 98.75 | 84.73 | 64.52 | 99.71 | 6.60 | 6.18 |
| | SalUn | 87.91 | 99.74 | 75.72 | 100.00 | 3.20 | 6.21 |
| | Unlink | 98.07 | 97.44 | 75.17 | 100.00 | **1.35** | **0.004** |
| | Retrain | 100.00 | 100.00 | 95.95 | 100.00 | - | 118.44 |
| | FT | 100.00 | 96.94 | 93.23 | 100.00 | 1.44 | 7.41 |
| | GA | 97.39 | 98.07 | 94.24 | 98.93 | 1.56 | 0.43 |
| | IU | 86.12 | 95.32 | 91.71 | 98.42 | 6.09 | 1.23 |
| SVHN | BE | 99.99 | 98.41 | 94.08 | 100.00 | 0.87 | 0.98 |
| | BS | 90.40 | 99.42 | 95.59 | 99.85 | 2.66 | 2.09 |
| | $\ell_1$-sparse | 100.00 | 98.34 | 94.38 | 100.00 | 0.80 | 7.60 |
| | SalUn | 99.99 | 99.99 | 96.36 | 100.00 | **0.11** | 8.21 |
| | Unlink | 97.36 | 99.40 | 95.92 | 100.00 | 0.81 | **0.04** |

Table 17: Results of class-wise forgetting on VGG16.

| Dataset | Methods | UA↑ | RA↑ | TA↑ | MIA↑ | Avg.Gap↓ | RTE (Mins)↓ |
|---|---|---|---|---|---|---|---|
| | Retrain | 100.00 | 99.99 | 93.69 | 100.00 | - | 27.74 |
| | FT | 100.00 | 93.46 | 87.44 | 100.00 | 3.19 | 1.74 |
| | GA | 99.81 | 93.23 | 86.58 | 99.89 | 3.54 | 0.12 |
| | IU | 82.22 | 96.93 | 63.24 | 88.86 | 11.73 | 0.36 |
| CIFAR-10 | BE | 98.70 | 95.54 | 87.92 | 99.80 | 2.92 | 0.22 |
| | BS | 83.59 | 92.48 | 84.93 | 87.21 | 11.37 | 0.31 |
| | $\ell_1$-sparse | 99.03 | 97.17 | 90.69 | 100.00 | 1.48 | 1.76 |
| | SalUn | 100.00 | 98.19 | 91.69 | 100.00 | **0.95** | 1.90 |
| | Unlink | 95.65 | 99.38 | 93.69 | 100.00 | 1.23 | **0.015** |
| | Retrain | 100.00 | 98.64 | 69.58 | 100.00 | - | 30.76 |
| | FT | 74.67 | 94.94 | 67.64 | 91.58 | 9.85 | 1.89 |
| | GA | 100.00 | 88.42 | 63.33 | 100.00 | 4.12 | 0.03 |
| | IU | 82.22 | 86.94 | 63.24 | 88.86 | 11.73 | 0.36 |
| CIFAR-100 | BE | 88.11 | 88.39 | 63.42 | 91.69 | 9.15 | 0.04 |
| | BS | 83.11 | 89.23 | 64.01 | 88.27 | 10.90 | 0.05 |
| | $\ell_1$-sparse | 80.51 | 93.90 | 67.23 | 93.34 | 8.31 | 1.95 |
| | SalUn | 81.87 | 97.56 | 68.99 | 100.00 | 4.95 | 2.02 |
| | Unlink | 98.21 | 96.39 | 69.67 | 100.00 | **1.01** | **0.004** |
| | Retrain | 100.00 | 100.00 | 95.83 | 100.00 | - | 28.77 |
| | FT | 100.00 | 97.83 | 93.30 | 100.00 | 1.17 | 1.80 |
| | GA | 100.00 | 77.66 | 74.89 | 80.00 | 15.82 | 0.11 |
| | IU | 96.62 | 91.54 | 87.22 | 99.93 | 5.13 | 0.33 |

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

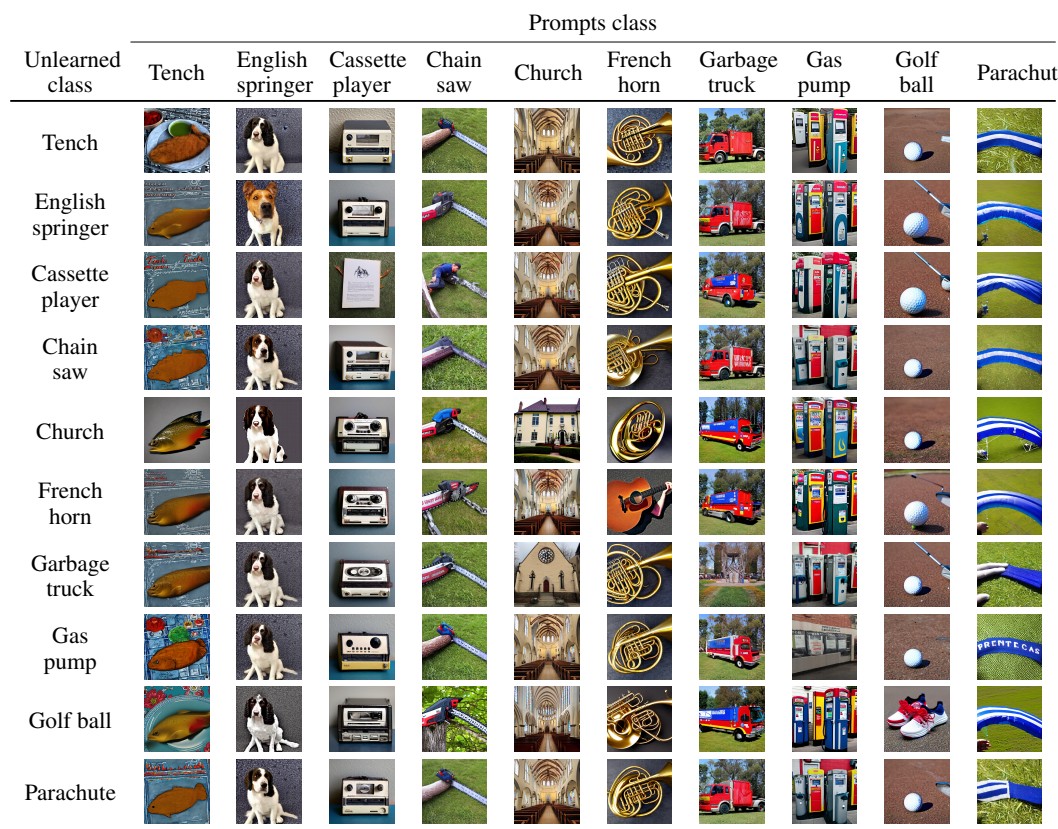

Figure 4: Visulalization of generated images by SD for class-wise forgetting on Imagenette.