# OpenReview forum: "Unlearn In a Blink"
_ICLR.cc/2026/Conference — ICLR 2026 Conference Withdrawn Submission_

### Official Review · Reviewer_D19q · 2025-10-23

**Soundness:** 2
**Presentation:** 2
**Contribution:** 2
**Rating:** 4
**Confidence:** 4

**Summary:**

The paper proposes a training-free MU method, Unlink. The core idea is to use only a few unlabeled samples from the forget set to extract features and perform SVD, obtaining a forgetting subspace. Selected layer weights are then projected and stripped of their components on this subspace, so forward activations related to the forgotten concept are suppressed while preserving performance on the remaining data. The method is instantiated for FC, MHSA, and convolutional layers. To handle strongly entangled remaining/forgetting features, the paper introduces a generalized Rayleigh-quotient formulation to find directions that jointly preserve remaining and suppress forgetting, weighting basis vectors by eigenvalues to balance forgetting and retention. Experiments on classification, vision–language, and diffusion generation report performance close to retraining-based upper bounds while being orders of magnitude faster.

**Strengths:**

1. The training-free and remaining-data-free setting directly addresses a practical need for constructing fast, lightweight MU baselines that are easier to deploy in engineering contexts.

2. The paper systematizes subspace-projection unlearning across multiple modules (FC/MHSA/Conv) without relying on retain data and introduces a generalized Rayleigh quotient to adaptively weight subspaces under high entanglement. The design is intuitive and compatible with modern architectures.

3. The method demonstrates strong speed and quality trade-offs across diverse tasks (classification, SD concept/class forgetting, CLIP), outperforming or matching several representative baselines (SalUn, ESD, FM-N, SSD, GF) while being substantially faster.

**Weaknesses:**

1. Orthogonal decomposition and projection ideas have been explored in unlearning (e.g., CURE[a], MRP[b]), weakening Unlink's contribution. The paper should explicitly articulate the motivation and procedural differences, and clearly indicate where Unlink’s novelty lies relative to other work.

2. **My primary concern lies in layer selection and applicability**. The paper offers insufficient detail on where and how Unlink is applied. The main text mentions only that “(class-wise forgetting) We apply our method to the last layer of model,” but it does not alter features produced before the last layer for the forget data. This can make Unlink appear effective while remaining vulnerable to adaptive adversaries: even after removing the last-layer parameters, applying a k-NN classifier to the inputs of the last layer may still yield strong classification performance, effectively defeating class-wise unlearning. The authors should investigate and explicitly acknowledge this potential risk, for example, by visualizing the features at the inputs of the Unlink-edited layer to assess separability and by reporting k-NN classification results there. In addition, the appendix only provides a single-layer ablation on VGG16. It remains inadequate for deeper stacks such as ResNet/ViT/CLIP/SD regarding which layers/heads/channels to edit and what general selection principles apply. It is currently unclear how the edited layers/heads/channels are chosen and how practitioners should select them on other architectures.

3. Table 3 shows limited retention under highly entangled features. For Unlink$^\dagger$, class 83 shows improvements over other methods but remains far below categories outside its super-class, which is insufficient for some applications and indicates limited retention when features are highly entangled.

### Reference
a. Biswas, Shristi Das et al. “CURE: Concept Unlearning via Orthogonal Representation Editing in Diffusion Models.” ArXiv.

b. Wu, Chengcan et al. “Reliable Unlearning Harmful Information in LLMs with Metamorphosis Representation Projection.” ArXiv.

**Questions:**

1. Clarify how Unlink differs from prior orthogonal projection approaches conceptually and procedurally.

2. For each setting and architecture in experiements, which specific layers/heads/channels are edited, and how are they chosen? Please provide ablation studies and, if possible, a general selection heuristic or automated rule.

3. Why is RTE identical for 1-shot and 5-shot in Table 6 (appendix)? I understand that using more forget samples should increase costs for feature extraction and matrix calculations.

---

### Official Review · Reviewer_U2Gz · 2025-10-25

**Soundness:** 3
**Presentation:** 3
**Contribution:** 3
**Rating:** 4
**Confidence:** 5

**Summary:**

The paper proposes Unlink (Unlearn in a Blink), a training-free and remaining-data-free machine unlearning approach. It removes targeted knowledge by directly editing model weights instead of retraining. The method estimates a low-dimensional subspace corresponding to the data to forget using SVD, then projects model weights orthogonally to that subspace (W* = W - WUfUf^T), effectively suppressing activations related to the forgotten concept while preserving others.

To handle overlapping features, the authors introduce an extended version Unlink† based on a generalized Rayleigh quotient, which balances forgetting and retention. The method generalizes to fully connected, attention, and convolutional layers, and applies to classification, image generation, and vision-language models.

Experiments show that Unlink achieves comparable or better unlearning quality than prior methods such as SalUn while being orders of magnitude faster (for example, 0.6 seconds versus more than two hours). It requires no retraining, no access to remaining data, and minimal hyperparameter tuning, making it a fast and practical baseline for future unlearning research.

**Strengths:**

1. This paper proposes a training-free and remaining-data-free unlearning method, which is quite novel.

2. The mathematical notation is clear and well-formulated, making the method easy to follow and understand.

**Weaknesses:**

1. The experiments in this paper mainly focus on the class-wise unlearning setting. In Table 10, only the 10% random forgetting case is presented, where Unlink already performs comparably to SalUn and ℓ₁-sparse. It would be valuable if the authors could include results with larger forgetting ratios, such as 20%, 30%, and 40% random forgetting, to better demonstrate the scalability and robustness of Unlink.

2. Sequential unlearning is also an important and realistic scenario worth investigating. For example, suppose we have four subsets of data and perform unlearning iteratively by forgetting 10% of the data each time, for a total of four rounds. I am curious to see how Unlink performs under this sequential unlearning setting.

**Questions:**

See weakness.

---

### Official Review · Reviewer_PiBH · 2025-10-29

**Soundness:** 3
**Presentation:** 2
**Contribution:** 1
**Rating:** 2
**Confidence:** 4

**Summary:**

The paper introduces Unlearn In a Blink (Unlink), a training-free and remaining-data-free baseline for Machine Unlearning (MU) across classification, generation, and multimodal vision-language tasks. The key idea is to remove low-dimensional subspaces associated with targeted (forgetting) concepts directly from the model’s weight space, effectively rendering the model “blind” to undesired content. Unlink computes the forgetting subspace via Singular Value Decomposition (SVD) on a small number of forgetting samples and removes the corresponding projection from the model weights.

**Strengths:**

1. Method Generability: The method is shown to be useful in diverse vision tasks such as classification, generation, and vision-language.
2. Hyperparameter Robustness: Hyperparameter sensitivity study shows performance remains strong with minimal hyperparameter tuning.
3. Theoretical Clarity: Solid mathematical derivation of their method and motivation is shown.

**Weaknesses:**

1. Tables 4 and 5 compare Unlink to several diffusion unlearning methods, including FMN, ESD, and SalUN. All of these rely on fine-tuning. While the comparisons are fair and the authors correctly note that SalUN achieves stronger unlearning while Unlink shows better FID retention and lower computational cost, the paper does not include ANY comparisons to training-free approaches. Methods such as UCE [1], ConceptPrune [2], and AdaVD [3] are all training-free and data-free, which are the same advantages that Unlink claims. Including these baselines would provide a more balanced and meaningful comparison to existing work in the same category.
2. Line 86 mentions that the method generalizes to various model architectures. The results on different vision tasks are promising, but it would be helpful to test the method on a broader range of image generation models. For example, Stable Diffusion v3 uses a Diffusion Transformer (DiT) instead of a U-Net. Evaluating Unlink on architectures like this would strengthen the claim of generalizability and show how the approach adapts to newer model families.

[1] Gandikota, Rohit, et al. "Unified concept editing in diffusion models." Proceedings of the IEEE/CVF Winter Conference on Applications of Computer Vision. 2024.

[2] Wang, Yuan, et al. "Precise, fast, and low-cost concept erasure in value space: Orthogonal complement matters." 2025 IEEE/CVF Conference on Computer Vision and Pattern Recognition (CVPR). IEEE, 2025.

[3] Chavhan, Ruchika, Da Li, and Timothy Hospedales. "Conceptprune: Concept editing in diffusion models via skilled neuron pruning." arXiv preprint arXiv:2405.19237 (2024).

**Questions:**

1. While the authors note that their approach differs from Deep Unlearning [4] by not requiring access to a retained dataset and being applicable beyond classification, the distinction remains somewhat underexplained. It would strengthen the paper to explicitly clarify how Unlink diverges from Deep Unlearning either mathematically or procedurally. My current understanding is that Deep Unlearning operates primarily in activation space, whereas Unlink performs subspace removal in weight space. Providing a clear rationale for differing design choices would help highlight the novelty and motivation of the proposed approach.


[4] Kodge, Sangamesh, Gobinda Saha, and Kaushik Roy. "Deep Unlearning: Fast and Efficient Gradient-free Approach to Class Forgetting." arXiv preprint arXiv:2312.00761 (2023).

---

### Official Review · Reviewer_uybC · 2025-11-01

**Soundness:** 2
**Presentation:** 2
**Contribution:** 2
**Rating:** 4
**Confidence:** 5

**Summary:**

This paper proposes Unlink, a novel machine forgetting method that requires no additional training or access to a preserving dataset. It eliminates the feature subspace associated with the content to be forgotten by directly modifying the model weights, thus making the model "blind" to that content. This process is so fast it can be completed "in the blink of an eye" and performs exceptionally well on various tasks, including classification, image generation, etc.

**Strengths:**

1. The motivation is clear and supportive.
2. The paper tackles an important problem and presents impressive speed benchmarks.
3. The compared methods are sufficient.

**Weaknesses:**

1. The primary method (Unlink) is based on the extremely strong and often invalid assumption that the forgetting and remaining concept subspaces are orthogonal. The authors are clearly aware that this assumption fails in practice, as evidenced by Table 3, where the main "Unlink" method completely erases an entangled class (class 83 accuracy drops to 0.00%). There are entangled classes in practice. Can the author conduct their experiments on face recognition tasks (in GS-LoRA++ [3]) and compare with their method to support the effectiveness of the proposed method?
2. Unrealistic Assumptions and Hand-wavy Theory: The theoretical justification (Section 3.1) relies on shaky ground.
The assumption that feature representations xf can be perfectly decomposed into a subspace component zf and a negligible residual ϵf is an oversimplification that does not hold in practice. The entire derivation in Eq. (5) and (7) hinges on this residual being small, but provides no bounds or analysis of what happens when it is not.
The "Neural Collapse" and "Tunnel Effect" literature is cited to justify subspace orthogonality. While these phenomena describe the collapse of class means, they do not guarantee the orthogonality of the entire feature subspaces, especially for fine-grained or semantically overlapping classes—the very setting where unlearning is most challenging. The paper's own experiments (Table 3) prove this assumption is false.
3. What is the real significance of using SVD? When performing classification tasks, we only need to modify the weights of the corresponding classes in the FFN layer (for example, by randomly initializing them) to achieve a complete forgetting without harming the performance of other retained classes.
4. The Stable Diffusion experiments are critically underspecified. How are the "few samples" for creating the forgetting subspace obtained for abstract concepts like "nudity"? Are they generated images? Text embeddings?
5. Insufficient Ablation: Why only apply the method to the last layer?
6. Membership inference attacks on a given model demonstrated that latent feature representations can leak information on whether individual data is used in training the model. Moreover, recent reconstruction attacks [1,2] successfully recover the data “forgotten” by the unlearned models, thereby exposing the risk of shallow unlearning by many existing approaches. The author should try this kind of attack.

[1] Reconstruction attacks on machine unlearning: Simple models are vulnerable

[2] Learn what you want to unlearn: Unlearning inversion attacks against machine unlearning

[3] Practical Continual Forgetting for Pre-trained Vision Models

**Questions:**

How do you calculate MIA? Can you explain it in details? Thanks

---

### Note · Authors · 2025-11-14

**Comment:**

We sincerely thank the reviewers for their time and feedback.

**Withdrawal Confirmation:**

I have read and agree with the venue's withdrawal policy on behalf of myself and my co-authors.